# Discovering Dynamic Salient Regions for Spatio-Temporal Graph Neural Networks

**Iulia Duta**[*]
Bitdefender, Romania
id366@cam.ac.uk

**Andrei Nicolicioiu**[*]
Bitdefender, Romania
anicolicioiu@bitdefender.com

**Marius Leordeanu**
Bitdefender, Romania
Institute of Mathematics of the Romanian Academy
University "Politehnica" of Bucharest
marius.leordeanu@imar.ro

## Abstract

Graph Neural Networks are perfectly suited to capture latent interactions between various entities in the spatio-temporal domain (e.g. videos). However, when an explicit structure is not available, it is not obvious what atomic elements should be represented as nodes. Current works generally use pre-trained object detectors or fixed, predefined regions to extract graph nodes. Improving upon this, our proposed model learns nodes that dynamically attach to well-delimited salient regions, which are relevant for a higher-level task, without using any object-level supervision. Constructing these localized, adaptive nodes gives our model inductive bias towards object-centric representations and we show that it discovers regions that are well correlated with objects in the video. In extensive ablation studies and experiments on two challenging datasets, we show superior performance to previous graph neural networks models for video classification.

## 1 Introduction

Spatio-temporal data, and videos, in particular, are characterised by an abundance of events that require complex reasoning to be understood. In such data, entities or classes exist at multiple scales and in different contexts in space and time, starting from lower-level physical objects, which are well localized in space and moving towards higher-level concepts which define complex interactions. We need a representation that captures such spatio-temporal interactions at different level of granularity, depending on the current scene and the requirements of the task. Classical convolutional nets address spatio-temporal processing in a simple and rigid manner, determined only by fixed local receptive fields [1]. Alternatively, space-time graph neural nets [2, 3] offer a more powerful and flexible approach modeling complex short and long-range interactions between visual entities.

In this paper, we propose a novel method to enhance vision Graph Neural Networks (GNNs) by an additional capability, missing from any other previous works. That is, to have nodes that are constructed for spatial reasoning and can adapt to the current input. Prior works are limited to having either nodes attached to semantic attention maps [4] or attached to fixed locations such as grids [5, 3, 6]. Moreover, unlike works that require external object detectors [7] our method relies on a learnable mechanism to adapt to the current input.

---

[*]Equal contribution.

35th Conference on Neural Information Processing Systems (NeurIPS 2021).

We propose a method that learns to discover salient regions, well-delimited in space and time, that are useful for modeling interactions between various entities. Such entities could be single objects, parts or groups of objects that perform together a simple action. Each node learns to associate by itself to such salient regions, thus the message passing between nodes is able to model object interactions more effectively. For humans, representing objects is a core knowledge system [8] and to emphasize them in our model, we predict salient regions [9] that give a strong inductive bias towards modeling them.

Our method, Dynamic Salient Regions Graph Neural Network (**DyReg-GNN**) improves the relational processing of videos by learning to discover salient regions that are relevant for the current scene and task. Note that the model learns to predict regions only from the weak supervision given by the high-level video classification loss, without supervision at the region level. Our experiments convincingly show that the regions discovered are well correlated with the objects present in the video, confirming the intuition that action recognition should be strongly related to salient region discovery. The capacity to discover such regions makes DyReg-GNN an excellent candidate model for tackling tasks requiring spatio-temporal reasoning.

**Our main contributions** are summarised as follow:

1. We propose a novel method to **augment spatio-temporal GNNs** by an additional capability: that of learning to create localized nodes suited for spatial reasoning, that adapt to the input.
2. The salient regions discovery **enhance the relational processing** for high-level video classification tasks: creating GNN nodes from predicted regions obtains superior performance compared to both using pre-trained object detectors or fixed regions
3. Our model leads to **unsupervised salient regions discovery**, a novelty in the realm of GNNs: it predicts such regions in videos, with only weak supervision at the video class level. We show that regions discovered are well correlated with actual physical object instances.

## 2 Related work

**Graph Neural Networks in Vision.** GNNs have been recently used in many domains where the data has a non-uniform structure [10, 11, 12, 13]. In vision tasks, it is important to model the relations between different entities appearing in the scene [14, 15] and GNNs have strong inductive biases towards relations [16, 17], thus they are perfectly suited for modeling interactions between visual instances. Since an explicit structure is not available in the video, it is of critical importance to establish what atomic elements should be represented as graph nodes. As our main contribution revolves around the creation of nodes, we analyse other recent GNN methods regarding the type of information that each node represents, and group them into two categories, *semantic* and *spatial*.

The approaches of [4, 18, 19, 20, 21, 22] capture the purely *semantic* interactions by reasoning over global graph nodes, each one receiving information from all the points in the input, regardless of spatio-temporal position. In [4] the nodes assignments are predicted from the input, while in [18] the associations between input and nodes are made by a soft clusterization. The work of [22] discovers different representation groups by using an iterative clusterization based on self-attention similarity.

The downside of these semantic approaches is that individual instances, especially those belonging to the same category, are not distinguished in the graph processing. This information is essential in tasks such as capturing human-object interactions, instance segmentation or tracking.

Alternatively, multiple methods, including ours, favour modeling instance interactions by defining *spatial* nodes associated with certain locations. We distinguish between them by how they extract the nodes from spatial location: as fixed regions or points [23, 24], or detected object boxes [25, 26, 27, 28, 29]. The method [5] creates nodes from every point in 2D convolutional features maps, while Non-Local [30] uses self-attention [31] between all spatio-temporal positions to capture distant interactions. Further, [3] extract nodes from larger fixed regions at different scales and processes them recurrently. Recent methods based on Transformer [32, 6, 33] also model the interactions between fixed locations on a grid using self-attention. In [7], nodes are created from object boxes extracted by an external detector and are processed using two different graph structures, one given by location and one given by nodes similarity. A related approach is used in [27] in a streaming setting while [34] learns to hop over unnecessary frames. Hybrid approaches use nodes corresponding to points and object features [35, 36] or propagate over both semantic and spatial nodes [37, 38, 39].

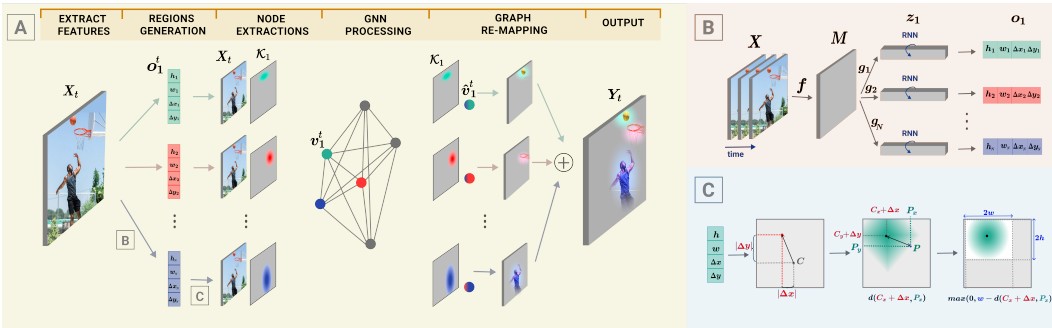

Figure 1: (**Left**) DyReg-GNN extracts localized node useful for relational processing of videos. For each node $i$, from the features $X_t$, we predict params $\mathbf{o}_i$ denoting the location and size of a region. They define a kernel $K_i$, used to extract the localized features $\mathbf{v}_i$ from the corresponding region of $X_t$. We process the nodes with a spatio-temporal GNN and project each node $\hat{\mathbf{v}}_i$ into its initial location. (**Right**) B) Node Region Generation: Functions $f$ and $\{g_i\}$ generate the regions params $\mathbf{o}_i$; $f$ extracts a latent representation shared between nodes, while each $g_i$ has different params for each node $i$. C) Node Features Extraction: Each $\mathbf{o}_i$ creates a kernel that is used in a differentiable pooling w.r.t. $\mathbf{o}_i$. This allows us to optimize the generation of these regions' params from the final classification loss, resulting in an unsupervised discovery of salient regions.

However, methods that rely on external modules trained on additional data, such as object detectors, are too dependent on the module's performance. They are unable to adapt to the current problem, being limited to the set of pre-defined annotations designed for another task. Differently, our module is optimized to discover regions useful for the current task, using only the video classification signal.

Recently, the method [40] uses multiple position-aware nodes that take into account the spatial structure. This makes it more suitable for capturing instances, but the nodes have associated a static learned location, where each one is biased towards a specific position regardless of the input. On the other hand, we dynamically assign a location for each node, based on the input, making the method more flexible to adapt to new scenes.

**Dynamic Networks.** Several works use second-order computations by dynamically predicting different parts of their model from the input, instead of directly optimising parameters. Our work is related to STN [41] that aggregates features by interpolating from an area given by a predicted global transformation and to the differentiable pooling used in some object detectors [42, 43, 44]. The method [45], replaces the parameters in a standard convolution with weights predicted from the input, resulting in a dynamically generated filter. Deformable convolutions [46, 47] predict, based on the input, an offset for each position in the convolutional kernel. Similar, [48] use the same idea of predicting offsets but in a graph formulation. The common topic of these methods is to predict dynamically a support for all points in a convolutional operation while we dynamically generate the input for a set of nodes designed to process high-level interactions. Related ideas, involving high-level processing of a small set of powerful modules, is also highlighted in [49] and [40].

**Unsupervised Object Representations.** There is an entire area of work devoted to extracting representations centered on objects [50] in a fully unsupervised setting [51, 52, 53, 22]. They are successful in leveraging a reconstruction task to decompose the scene into objects, for synthetic images. In [54] it is shown that representations learned from unsupervised decomposition are also helpful in relational reasoning tasks. Methods for generating unsupervised keypoints or entities [55, 56, 57, 58] have been generally used in synthetic setting. The method [55] generates keypoints from real images of people and faces but they use an image reconstruction objective that could not be aligned with the downsteam task. Our goal is to relate spatio-temporal entities, but without enforcing a clear decomposition of the scene into objects. This allows us to use a simpler but effective method that learns from classification supervision of real-world videos and obtain representations that are correlated to objects.

**Activity Recognition.** Video classification has been influenced by methods designed for 2D images [59, 60, 61, 62]. More powerful 3D convolutional networks have been later proposed [63], while

other methods factorise the 3D convolutions [64, 65, 66] bringing both computational speed and accuracy. Methods like TSM [67] and [68] showed that a simple shift in the convolutional features results in improved accuracy at a low computational budget.

## 3 Dynamic Salient Regions GNNs

We investigate how to create node representations that are useful for modeling visual interactions between various entities in space and time using GNNs. Our proposed Dynamic Salient Regions GNN model (DyReg-GNN) learns to dynamically assign each node to a certain interesting region. By dynamic, we mean that we have a fixed number $N$ of regions that change their position and size according to the input at each time step. The regions assigned to each of the $N$ nodes can change from one moment of time to the next depending on their saliency.

The main architecture of our DyReg-GNN model is illustrated in Figure 1. Our model receives feature volume $X \in \mathbb{R}^{T \times H \times W \times C}$ and at each time step $t$ we predict the location and size of $N$ regions. From these regions, a differentiable pooling operation creates graph nodes that are processed by a GNN and then are projected to their initial position. This module can be inserted at any intermediate level in a standard convolutional model.

### 3.1 Node Region Generation

We want to attend only to a few most relevant entities in the scene, thus a small number of nodes are used in DyReg-GNN (in our experiments $N = 9$) and it is crucial to assign them to the most salient regions. The number of nodes is a hyperparameter that we choose such that it exceeds the expected number of relevant entities in the scene, to increase the robustness of the model. Thus, we propose a global processing (shown in Figure 1 B) that aggregates the entire input features to produce regions defined by parameters indicating their location $(\Delta x, \Delta y)$ and size $(w, h)$.

To generate $N$ salient regions, we process the input $X_t$ using position-aware functions $f$ and $\{g_i\}_{i \in \overline{1,N}}$ that retain spatial information. Nodes should be consistent across time, thus we generate their regions in the same way at all time steps, by sharing in time the parameters of $f$ and $\{g_i\}$. The function $f$ is a convolutional network that highlights the important regions from the input.

$$M_t = f(X_t) \in \mathbb{R}^{H' \times W' \times C'} \tag{1}$$

For each node $i$, we generate a latent representation of its associated region using the $\{g_i\}$ functions. Each $g_i$ has the same architecture, but different parameters for each node and could be instantiated as a fully connected network or as global pooling enriched with spatial positional information. We generate the node regions from a global view to make the decision as informed as possible.

$$\hat{\mathbf{m}}_{i,t} = g_i(M_t) \in \mathbb{R}^{C'}, \forall i \in \overline{1,N} \tag{2}$$

Each of the $N$ latent representations is processed independently, with a GRU [69] recurrent network (shared between nodes), to take into account the past regions' representations.

$$\mathbf{z}_{i,t} = \text{GRU}(\mathbf{z}_{i,t-1}, \hat{\mathbf{m}}_{i,t}) \in \mathbb{R}^{C'}, \forall i \in \overline{1,N} \tag{3}$$

At each time step, the final parameters are obtained by a linear projection $W_o \in \mathbb{R}^{C' \times 4}$, transformed by a function $\alpha$ to control the initialisation of the position and size (e.g. regions would start at reference points either in the center of the frame or arranged on a grid). For more details about how to set the transformation $\alpha$ we refer to the Supplemental Materials.

$$\mathbf{o}_{i,t} = (\Delta x_{i,t}, \Delta y_{i,t}, w_{i,t}, h_{i,t}) = \alpha(W_o \mathbf{z}_{i,t}) \in \mathbb{R}^4 \tag{4}$$

### 3.2 Node Features Extraction

The following operations are applied independently at each time step thus, in the current subsection, we ignore the time index for clarity. We extract the features corresponding to each region $i$ using a differentiable pooling w.r.t. the predicted region parameters $\mathbf{o}_i$. All the input spatial locations $p \in \mathbb{R}^2$ are interpolated according to the kernel function $\mathcal{K}^{(i)}(p)$ as presented in Figure 1 C.

We present the operation for a single axis since the kernel is separable, acting in the same way on both axes:

$$\mathcal{K}^{(i)}(p_x, p_y) = k_x^{(i)}(p_x)k_y^{(i)}(p_y) \in \mathbb{R} \tag{5}$$

We define the center of the estimated region $c_{i,x} + \Delta x_i$, where $c_{i,x}$ is a fixed reference point for node $i$ (located in the frame's center or arranged on a grid). The values of the kernel decrease with the distance to the center and is non-zero up to a maximal distance of $w_i$, where $w_i$ and $\Delta x_i$ are the predicted parameters from Eq. 4.

$$k_x^{(i)}(p_x) = \max(0, w_i - |c_{i,x} + \Delta x_i - p_x|) \tag{6}$$

For each time step $t$, node $i$ is created by interpolating all points in the input $X_t$ using the kernel function. By modifying $(\Delta x_i, \Delta y_i)$ the network controls the location of the regions, while $(h_i, w_i)$ parameters indicate their size.

$$\mathbf{v}_i = \sum_{p_x=1}^{W} \sum_{p_y=1}^{H} \mathcal{K}^{(i)}(p_x, p_y)\mathbf{x}_{p_x, p_y} \in \mathbb{R}^C \tag{7}$$

Setting $w_i = 1$ leads to standard bilinear interpolation, but optimising it allows the model to adapt region's size and we observe that larger ones result in a more stable optimisation (see node size ablations from Supp. Material).

The position of the region associated with each node should be taken into account. It helps the relational processing by providing an identity for the node and is also useful in tasks that require positional information. We achieve this by computing a positional embedding for each node $i$ using a linear projection of the kernel $\mathcal{K}_i$ into the same space as the feature vector $v_i$ and summing them.

**Key Properties.** By construction, the nodes in our method are *localized*, meaning that they are clearly associated with a location: they pool information from clearly delimited area in space and they maintain position information from the positional embedding. These two aspects could be helpful in tasks involving spatio-temporal reasoning.

The *dynamic* aspect refers to the key capability of adapting the region's position and size according to the saliency of the input at each time step. This is done by predicting the regions from the input with the operations from equations (1–4).

An essential aspect of this method is that the final classification loss is *differentiable* with respect to regions' parameters as the gradients are passing from the nodes outputs $v_i$ through the kernels $k_i$ to the parameters $w_i$ and $\Delta x_i$. This allows us to learn regions from the final loss, *without direct supervision for the region generation*. Thus the method has more flexibility in learning relevant regions as appropriate for the task.

### 3.3 Graph Processing

For processing the nodes' features, different spatio-temporal GNNs could be used. Generally, they follow a framework [12] of sending messages between connected nodes, aggregating [70, 71] and updating them.

The specific message-passing mechanism is not the focus of the current work, thus we follow a general formulation similar to [3] for recurrent spatio-temporal graph processing. It uses two different stages: one happening between all the nodes at a single time step and the other one updating each node across time. For each time step $t$, we send messages between each pair of two nodes, computed as an MLP (with shared parameters) and aggregates them using a dot product attention $a(v_i, v_j) \in \mathbb{R}$.

$$\mathbf{v}_{i,t} = \sum_{j=1}^{N} a(\mathbf{v}_{j,t}, \mathbf{v}_{i,t})\text{MLP}([\mathbf{v}_{j,t}; \mathbf{v}_{i,t}]) \in \mathbb{R}^C \tag{8}$$

We incorporate temporal information through a shared recurrent function across time, applied independently for each node.

$$\hat{\mathbf{v}}_{i,t+1} = \text{GRU}(\hat{\mathbf{v}}_{i,t}, \mathbf{v}_{i,t}) \in \mathbb{R}^C \tag{9}$$

The GRU output represents the updated nodes' features and the two steps are repeated $K = 3$ times.

Table 1: Results on val. set of Smt-Smt-V2 showing the **importance of salient regions discovery.** We compare our predicted (unsupervised) regions to fixed grid regions or boxes given by an object detector using the same GNN model. The mean $L_2$ distance between the regions and gt. objects proves that DyReG-GNN has regions correlated with objects, while also having superior accuracy and efficiency.

| Model | Regions discovery | FLOPS $\downarrow$ | Dist $\downarrow$ | Acc (%)$\uparrow$ |
|---|---|---|---|---|
| TSM-R50 | - | 65.8G | - | 63.4 |
| + GNN+Fixed | Grid | +1.4G | 0.170 | 64.1 |
| + GNN+Detector | Obj detector | +41.1G | 0.125 | 64.0 |
| + DyReg-GNN | Unsupervised | +1.6G | 0.129 | **64.8** |

Table 2: **Consistent improvements over different backbones** on the validation set of Smt-Smt-V1 using central crop evaluation.

| Model | Acc (%) |
|---|---|
| TSM-R18 | 33.7 |
| TSM-R18 + DyReg-GNN | 35.6 ($\uparrow$ 1.9) |
| I3D-R50 | 44.0 |
| I3D-R50 + DyReg-GNN | 45.4 ($\uparrow$ 1.4) |
| TSM-R50 | 47.2 |
| TSM-R50 + DyReg-GNN | 48.8 ($\uparrow$ 1.6) |

## 3.4 Graph Re-Mapping

To use our method as a module inside any backbone, we produce an output with the same shape as the convolutional input $X_t \in \mathbb{R}^{H \times W \times C}$. The resulting features of each node are sent to all locations in the input according to the weights used in the initial pooling from Section 3.2.

$$\mathbf{y}_{p_x,p_y,t} = \sum_{i=1}^{N} \mathcal{K}_t^{(i)}(p_x, p_y)\hat{\mathbf{v}}_{i,t} \in \mathbb{R}^C \tag{10}$$

## 4 Experimental Analysis

While much effort is put into the creation of different video datasets used in the literature, such as Kinetics [63] or Charades [72], it has been argued [73] that they contain biases that make them solvable without complex spatio-temporal reasoning. CATER [73] is proposed to alleviate this, but it is too small (5500 videos) and still has biases that make the last few frames sufficient for good performance [34]. We test our model on two video classification datasets that seem to offer the best advantages, being large enough and requiring abilities to model complex interactions. We evaluate on real-world datasets, Something-Something-V1&V2 [74], while we also test on a variant of the SyncMNIST [3] dataset that is challenging and requires spatio-temporal reasoning, while allowing fast experimentation. The code for our method can be found in our repository [2].

### 4.1 Human-Object Interactions Experiments

Something-Something-V1&V2 [74] datasets classify scenes involving human-object complex interactions. They consist of 86K / 169K training videos and 11K / 25K validation videos, having 174 classes. Unless otherwise specified, all experiments on Something-Something datasets use TSM-ResNet-50 [67] as a backbone and we add instances of our module at multiple stages.

**Studying the Importance of Salient Regions Discovery.** We test the importance of the dynamic regions for GNNs vision methods by training models where we replace the predicted regions with the same number of fixed regions on a grid (GNN + Fixed Regions) or boxes (GNN + Detector) as given by a Faster R-CNN [75] trained on MSCOCO [76].

The detector based model has comparable results to the one with fixed regions, seemingly being unable to fully benefit from the correctly identified objects. The relative weaker performance of this model could be due to the fact that the pre-trained detector is not well aligned to the actual salient regions that are relevant for the classification problem.

On the other hand, this weakness is not applicable for DyReg-GNN that learns suitable regions for the current task and it obtains the best performance as seen in Table 1. Not only that it does not require object annotations, but it is also more computationally efficient. Running the detector on a video of

---

[2]https://github.com/bit-ml/DyReg-GNN

Table 3: **Results on val. set of Smt-Smt-V1.** Our model achieves competitive results compared to recent works (best results in red), while it outperforms all other graph-based methods (best results in blue). All the methods use ResNet50 as backbone.

|  | Model | Regions discovery | #F | Top 1 | Top 5 |
|---|---|---|---|---|---|
| non-Graph | TSM [67] | - | 16 | 48.4 | 78.1 |
| | S3D [64] | - | 64 | 48.2 | 78.7 |
| | GST [78] | - | 16 | 48.6 | 77.9 |
| | SmallBig [79] | - | 16 | 50.0 | 79.8 |
| | STM [80] | - | 16 | 50.7 | 80.4 |
| | MSNet [81] | - | 16 | 52.1 | 82.3 |
| Graph | ORN [14] | Objects | 8 | 36.0 | - |
| | NL I3D [7] | Grid | 32 | 44.4 | 76.0 |
| | NL GCN [7] | Objects | 32 | 46.1 | 76.8 |
| | TRG [82] | Frames | 16 | 48.1 | 80.4 |
| | RSTG [3] | Grid | 32 | 49.2 | 78.8 |
| | TSM+DyReg | Dynamic | 16 | 49.9 | 79.0 |

Table 4: **Results on val. set of Smt-Smt-V2.**, in comparisons to recent works. DyReg-GNN improves the TSM-ResNet50 backbone when using either one (r4) or three (r3-4-5) modules of graph processing and it obtains top results.

| Model | BB | Top 1 | Top 5 |
|---|---|---|---|
| TRG [82] | R50 | 59.8 | 87.4 |
| GST [78] | R50 | 62.6 | 87.9 |
| v-DP [83] | D121 | 62.9 | 88.0 |
| SmallBig [79] | R50 | 63.8 | 88.9 |
| STM [80] | R50 | 64.2 | 89.8 |
| MSNet [81] | R50 | 64.7 | 89.4 |
| TSM [67] | R50 | 63.4 | 88.5 |
| TSM+DyReg-r4 | R50 | 64.3 | 88.9 |
| TSM+DyReg-r3-4-5 | R50 | 64.8 | 89.4 |

size $224 \times 224$ would add $39.7$ GFLOPS on its own, comparing to the $1.6$G of three DyReg-GNN modules, from which $0.2$G represents the regions prediction.

Overall, our method, with unsupervised regions obtains superior performance in terms of accuracy and computational efficiency representing a suitable choice for relational processing of a video.

**Object-centric representations.** The nodes represent the core processing units and their localization enforces a clear decision on what specific regions to focus on while completely ignoring the rest, as a form of hard attention. Different from other works [77], our hard attention formulation is differentiable. To better understand what elements influence the model predictions, we could inspect the predicted kernels, thus introducing another layer of interpretability to the model, on top of the capabilities offered by the convolutional backbone. Visualisations of our nodes' regions reveal that generally, they cover the objects in the scene. For example, in the first row of Figure 3 the nodes are placed around the phone in the first frames and then separate into two groups, one for the phone one for the hand.

The localized nodes make our model capable of discovering salient regions, leading to object-centric node representations. We quantify this capacity by measuring the mean $L_2$ distance (normalised to the size of the input) between the predicted regions and ground-truth (gt.) objects given by [28]. The metric is completely defined in the Supp. Materials. We observed that the score improves during the learning process (it reaches $0.129$ starting from $0.201$), although the model is not optimized for this task. This suggests that the model actually learns object-centric representations.

In Table 1 we also compare the final $L_2$ distance of our best DyReg-GNN model to an object detector and to fixed grid regions. Although our method is not designed and supervised to find object regions, we observe that it is able to predict locations that are fairly close to gt. objects. The $L_2$ distance is similar to the one obtained by an external model ($0.129$ vs $0.125$), trained especially for detecting objects.

We observe that learning the regions' size is important for the stability of the optimisation and thus for the final performance (see Tab.5 and Supp. Material - Regions' Size section). However, the predicted size is not as well aligned with the size of the true objects. This gives us a hint that for the action classification task it is important to have good region locations, but their size is less relevant. We leave a more thoroughly investigation for futures work.

These experiments prove that the high-level classification task is well inter-related with the discovery of salient regions and that, in turn, these regions improve the relational processing in the recognition task. First, we show that DyReg-GNN's region obtain superior accuracy and efficiency than other methods of extracting nodes and second, these regions are well correlated to gt. object locations.

**Comparison to recent methods.** DyReg-GNN can be used with any convolutional model and we show that it consistently boosts the performance of multiple backbones(Table 2). We compare to recent methods from the literature in Table 3 and Table 4. Our method improves the accuracy over the

TSM-ResNet50 backbone on both Smt-Smt-V1 and Smt-Smt-V2 by $1.5\%$ and $1.4\%$ respectively and achieves competitive results. Compared to all the other graph based methods we obtain superior results, showing that our discovery of dynamic regions is effective for space-time relational processing.

**Implementation Details** Unless otherwise specified, we use TSM-ResNet50 (pre-trained on ImageNet [84]) as our backbone and add instances of our module in the last three stages. To benefit from ImageNet pre-training, we add our graph module as a residual connection. We noticed that models using multiple graphs have problems learning to adapt the regions from certain layers. We fix this by training models containing a single graph at each single considered stage, as the optimisation process is smoother for a single module, and distill their learned offsets into the bigger model. The distillation is done for the first $10\%$ of the training iterations to kick-start the optimization process and then continue the learning process using only the video classification signal.

In all experiments we follow the training setting of [67], using 16 frames resized to have the shorter side of size 256, and randomly sample a crop of size $224 \times 224$. For the evaluations, we follow the setting in [67] of taking 3 spatial crops of size $256 \times 256$ with 2 temporal samplings and averaging their results. For training, we use SGD optimizer with learning rate $0.001$ and momentum $0.9$, using a total batch-size of 10, trained on two GPUs. We decrease the learning rate by a factor of 10 three times when the optimisation reaches a plateau.

## 4.2 Synthetic Experiments

SyncMNIST is a synthetic dataset involving digits that move on a black background, some in a random manner, while some move synchronously. The task is to identify the digits that move in the same way. We use a harder variant of the dataset (MultiSyncMNIST), where the videos could include multiple digits of the same class. The challenge consists in finding useful entities and model their relationships while being able to distinguish between instances of the same class. Each video contain 5 digits and the goal is to find the smallest and the largest digit class among the subset that moves in the same way. This results in a video classification task with 56 classes. The dataset contains 600k training videos and 10k validation videos with 10 frames each.

**Studying the Importance of Dynamic Nodes.** We validate our assumption that the nodes should be dynamic, meaning that their regions position and size should be adapted according to the input at each time step. We investigate (Table. 5) different types of localized nodes, each adapting to the input to a varying degree, and show the benefits of our design choices. We experiment with variants of our model, all having the same backbone (2D ResNet-18 [85]), the same graph processing and same pre-determined number of regions, but we constrain the node regions in different ways.

*Fixed Model* extracts node features from regions arranged on a grid, with a fix location and size.

*Static Model* investigates the importance of dynamic regions by optimising regions based on the whole dataset but do not take into account the current input. Effectively, the features $\mathbf{z}_i$ from Eq. 4 become learnable parameters.

*Constant-Time Model* has regions adapted to the current video but they do not change in time.

*DyReg-GNN Model* predicts regions defined by location and size, and we can either pre-determine a fixed size for all the regions (*Position-Only Model*) or directly predict it from the input as in our complete model (*DyReg-GNN Model*).

These experiments (Table 5), show that the fixed region approach (Fixed Model) achieves the worst results, slightly improving when the regions are allowed to change according to the learned statistics of the dataset (Static model). Adapting to the input is shown to be beneficial, the performance improving even when the regions are invariant in time (Constant-Time Model), and further more when predicting different regions at every time steps (Position-Only). The best performance is achieved when both the location and the size of the regions are dynamically predicted from the input (DyReg-GNN).

In Figure 2 we show examples of the kernels obtained for each of these models. We observe that the Static Model's kernels are learned to be arranged uniformly on a grid, to cover all possible movements in the scene, while the Constant-Time Model's kernels are adapted for each video such that they cover the main area where the digits move in the current video. The full DyReg-GNN Model learns to reduce the size of its regions and we observe that they closely follow the movement of the digits.

The previous experiments show that performance increases when the model becomes more dynamic, proving that our model benefits from nodes that are adapted to a higher degree to the current input.

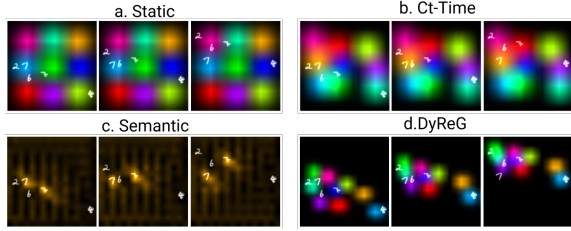

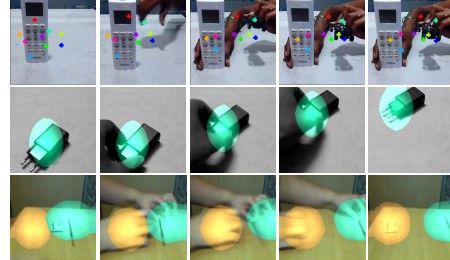

Figure 2: **Nodes' regions** on MultiSyncMNIST for 3 frames. a) *Static* Model, ignoring the input, learns a regular grid; b) *Constant-Time* predicts the same regions for all time steps, covering the movement in the video; c) The attention map of a single node of *Semantic* that can't distinguish between different instances of the same digit; d) *DyReg-GNN* generally follows the digits locations at each time steps while also adapting the regions' size.

Figure 3: **Nodes' regions** on Smt-Smt-V2. We show (1st row) the center of all the $N$ regions as predicted by DyReg-GNN (each color for a node). Each node region (last 2 rows) corresponds to a zone from the *latent* conv features pooled by a node.

Table 5: **Ablation of dynamic nodes on MultiSyncM-NIST.** It is crucial to have regions that adapt based on the input (Dynamic), both their position (Pos.) and size at each time step.

Table 6: **Semantic vs spatial nodes** on MultiSyncMNIST. The localized (spatial) node regions of DyReg-GNN are better suited than semantic nodes' maps obtained by the Semantic Model.

| Model | Optimise Pos. | Time Varying | Dynamic Pos. | Size | Acc |
|---|---|---|---|---|---|
| Fixed | | | | | 78.85 |
| Static | ✓ | | | | 81.48 |
| Ct-Time | ✓ | | ✓ | | 86.77 |
| Pos-Only | ✓ | ✓ | ✓ | | 93.41 |
| DyReg-GNN | ✓ | ✓ | ✓ | ✓ | **95.09** |

| Model | Params (M) | Acc |
|---|---|---|
| ResNet-18 | 2.79 | 52.29 |
| Fixed | 2.82 | 78.85 |
| Semantic | 2.85 | 82.41 |
| DyReg-GNN-Lite | 2.83 | 91.43 |
| DyReg-GNN | 3.08 | **95.09** |

**Studying the Importance of Localized Nodes.** We argue that nodes should pool information from different locations according to the input, such that the extracted features correspond to meaningful entities. Depending on the goal, we could balance between semantic nodes globally extracted from all spatial positions or localized (spatial) nodes that are obtained from well-delimited regions.

*Semantic Model* creates nodes similar to [4, 19] where each node extracts features from all the spatial locations and could represent a semantic concept. Each node is extracted by a global average pooling where the weights at every position $p$ are directly predicted from the input features at that location. Practically, we replace the spatially delimited kernel used in our model with this global attention map.

A major downside of this approach is that it does not distinguish between positions with the same features, making it harder to reason about different instances. Figure 2.C shows the attention map of a single node and we observe that it has equally high activations for both instances of the same digit, thus making it hard to distinguish between them.

This limitation does not exist in our DyReg-GNN model, as it predicts localized nodes that favour the modeling of instances. For comparison, we use two variants with a different number of parameters and show that they clearly outperform the semantic model (Table 6). These experiments prove that in cases that involve spatial reasoning of entities, such as the current task, DyReg-GNN is a perfect choice, showing its benefits for spatio-temporal modeling.

**Implementation details.** All models share the ResNet-18 backbone with 3 stages, where the graph receives the features from the second stage and sends its output to the third stage. We use $N = 9$ graph nodes and repeat the graph propagation for three iterations. In our main model, $f$ from Eq. 1 is a small convolutional network while $g$ is a fully connected layer. For the lighter model that implements $g$ as a global pooling enriched with spatial positional information, we refer to the Supp. Materials. The graph offsets are initialized such that all the nodes' regions start in the center of the frame. In all experiments, we use SGD optimizer with learning rate 0.001 and momentum 0.9, trained on a single GPU.

**Key Results.** In the previous section, we experimentally validated that: **1.** DyReg-GNN consistently improves multiple backbones (Table 2) obtaining competitive results (Table 3, 4); **2.** learned dynamic

regions are crucial for good performance (Table 5) and **3.** these regions are preferable to fixed regions or external object detectors for space-time GNNs (Table 1); **4.** predicted nodes correspond to salient regions (Fig. 2-3) and are well correlated with objects (Table 1).

## 5 Conclusions

We propose Dynamic Salient Regions Graph Neural Networks (DyReg-GNN), a relational model for processing spatio-temporal data (videos), that augments visual GNNs by learning to predict localized nodes, adapted for the current scene. This novel method enhances the relational processing of spatio-temporal GNNs and we experimentally prove that it is superior to having nodes anchored in fixed predefined regions or linked to external pre-trained object detectors. Although we do not use region level supervision, the learning dynamics of high-level classification produces salient regions that are well correlated with object instances. We believe that our method of learning dynamic, localized nodes is a valuable direction that could lead to further advances to the growing number of powerful relational models in spatio-temporal domains.

**Acknowledgment** We would like to thank Florin Brad, Elena Burceanu and Florin Gogianu for their valuable feedback and discussions of this work. This work has been supported in part by Bitdefender and UEFISCDI, through projects EEA-RO-2018-0496 and PN-III-P4-ID-PCE-2020-2819.

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
