In this Appendix we present an impact statement, discuss some limitations of the method and then we provide more technical details about DyReg-GNN model and include some additional visualisations and ablation studies.

Section A presents some views on the broader impact of this work.

Section B identifies some limitations of the methods.

Section C presents more details about how the regions are generated.

Section D shows a qualitative analysis of the regions predicted by our model.

Section E shows additional ablation studies in the synthetic setting in relation to the number of nodes, the regions size, the importance of recurrence when generating the nodes and comparisons to using ground-truth boxes or other baselines.

Section F presents some training details, describe the metric used to measure the correlation between our regions and the existing objects in the scene and have a runtime analysis of our proposed module.

We provide our full code as supplementary material and we will release it online upon the paper publication. Beside this Appendix, we also provide some videos, visualising the regions discovered by our DyReg-GNN model.

## A  Broader Impact

We research novel methods that would improve current general models for spatio-temporal processing. Our goal is to investigate models that emphasize a small number of relevant nodes having the potential to be more explainable and that could lead to more interpretable reasoning. Although this is not fully realised in this paper, we believe that this work is a good step in this direction. Our model enhances any convolutional backbone for video processing and thus inherits the benefits and also the possible harms brought by such models.

When developing our model, we used a synthetic dataset of moving digits and a public dataset for human-object interactions. Our model is kept generic, with no parts specially designed for these tasks. The models trained on these datasets have no obvious direct real application, as the first one is a toy dataset and the second one has restrictive classes meant only to evaluate the capabilities of the models. But developing better models for video understanding leads to more effective applications. On one hand, it could lead to better applications helping visually impaired people navigate the world and on the other hand it could lead to stricter automatic surveillance of workers. In order for ML technology to have a positive broader impact, more discussions between different actors in society should be conducted leading to the development of guidelines and practices.

The proposed work does not rely on using object detectors and only uses video level supervision. Object detectors have a predefined list of objects, that would not be sufficient for many practical cases leading to biases in the system. Moreover, this way we eliminate a possible source of biases coming from the object-level annotations.

## B  Limitations

By design, DyReg-GNN uses a fixed number of nodes, that we treat as a hyperparameter. This way the model is forced to produce the same number of regions regardless of the complexity of the scene. From simpler scenes, the model learns to group the nodes in overlapping regions, creating redundancy. On the other hand, more complex scenes have an increased number of relevant regions, tending to require distinct regions. This could lead to a discrepancy that would increase the difficulty of the optimisation process. Changes in scene's complexity could be also observed in a single video when the scene suffer major changes in time. For example when elements appear, disappear or are occluded from view, the number of regions predicted by the model remains the same and it is

harder to properly model all the elements. Ideally, we want a system that adapts to the complexity of the scene by dynamically predicting the number of nodes. This is a challenging task that requires additional investigations and we leave it for future work.

Preliminary experiments reveals that our method requires a relative large amounts of data to be properly trained. This seems not to be an issue for Something-Something dataset that has 80k-160k training videos, but could be an issue for smaller datasets. On MultiSyncMNIST we could train models with high accuracy on $10\%$ of the whole dataset of 600k videos. But when using only $1\%$ (6k videos) of the data, the predicted regions would not change during training. Given the size of the recent video dataset, this is not a big limitation.

## C  Node Region Generation

The goal of this sub-module is to generate the regions that correspond to salient zones in the input. We achieve this by processing the input globally with position-aware functions $f$ and $\{g_i\}$.

**Function $f$.**  We use $f$ function to aggregate local information from larger regions in the input while preserving sufficient positional information. The input $X_t \in \mathbb{R}^{H \times W \times C}$ is first projected into a lower dimension $C'$ since this representation should only encode saliency without the need to precisely model visual elements. Then we increase the receptive field by applying two conv layers, followed by a transposed conv and then a final conv layer. This results in a feature map $M_t = f(X_t) \in \mathbb{R}^{H' \times W' \times C'}$. Depending on the backbone and the stage where the graph is added $H, W$ have different values and we adapt the hyperparameters of the convolutional layers such that $H'$ and $W'$ are not smaller than 6. For example, in the synthetic experiments $f$ reduces the input from $\mathbb{R}^{16 \times 16 \times 32}$ to $\mathbb{R}^{7 \times 7 \times 16}$.

**Functions $\{g_i\}$.**  For each node $i$ we use $g_i$ to extract a global latent representation from which we predict the corresponding region parameters. We present two variant of $g_i$ function, a larger and more precise one and a smaller, more computational efficient one.

For the bigger one, we use a simple fully connected layer of size $C \times (H' * W' * C')$ that takes the whole $M_t$ and produces a vector of size $C$. This way $g_i$ could distinguish and model the spatial locations of the $H' \times W'$ grid.

The second approach consists in a weighted global average pooling for each node $i$. The weight associated to each location $p$ is predicted directly from the input $M_{t,p}$ by a $1 \times 1$ convolution. But this results in a translation-invariant function $g_i$ that losses the location information. We alleviate this by adding to each of the $H' \times W'$ location a positional embedding similar to the one used in [31]. This approach predicts regions of slightly poorer quality as the location information is not perfectly encoded in the positional embeddings. For a lighter model, such as the one presented in Table 6 of the main paper we could use the second approach for the $\{g_i\}$ functions and also skip the $f$ processing.

**Constraints**  Equation 4 in the main paper could be expended as:

$$\tilde{\mathbf{o}}_i = (\Delta\tilde{x}_i, \Delta\tilde{y}_i, \tilde{w}_i, \tilde{h}_i) = \gamma \odot W_o \mathbf{z}_i \in \mathbb{R}^4 \tag{11}$$
$$\mathbf{o}_i = \alpha(\tilde{\mathbf{o}}_i)$$

To constrain the model to predict valid image regions and also to start from regions with favourable position and size, we apply non-linear functions for each component $\mathbf{o}_i = \alpha(\tilde{\mathbf{o}}_i)$. We design the non-linearities such that $w_i, h_i > 0$ and $\Delta x_i + C_x \in [0, W]$ and , $\Delta y_i + C_y \in [0, H]$, where $C$ is a fixed reference point. In experiments, all nodes share the same constant $C$, representing the center of the image.

$$h = e^{\tilde{h}} h_{init} \qquad w = e^{\tilde{w}} w_{init} \tag{12}$$

$$\Delta x = \frac{W}{2} \tanh\left(\Delta\tilde{x} + (\frac{2C_x}{W} - 1)\right) + \frac{W}{2} - C_x \tag{13}$$

$$\Delta y = \frac{H}{2} \tanh\left(\Delta\tilde{y} + (\frac{2C_y}{H} - 1)\right) + \frac{H}{2} - C_y \tag{14}$$

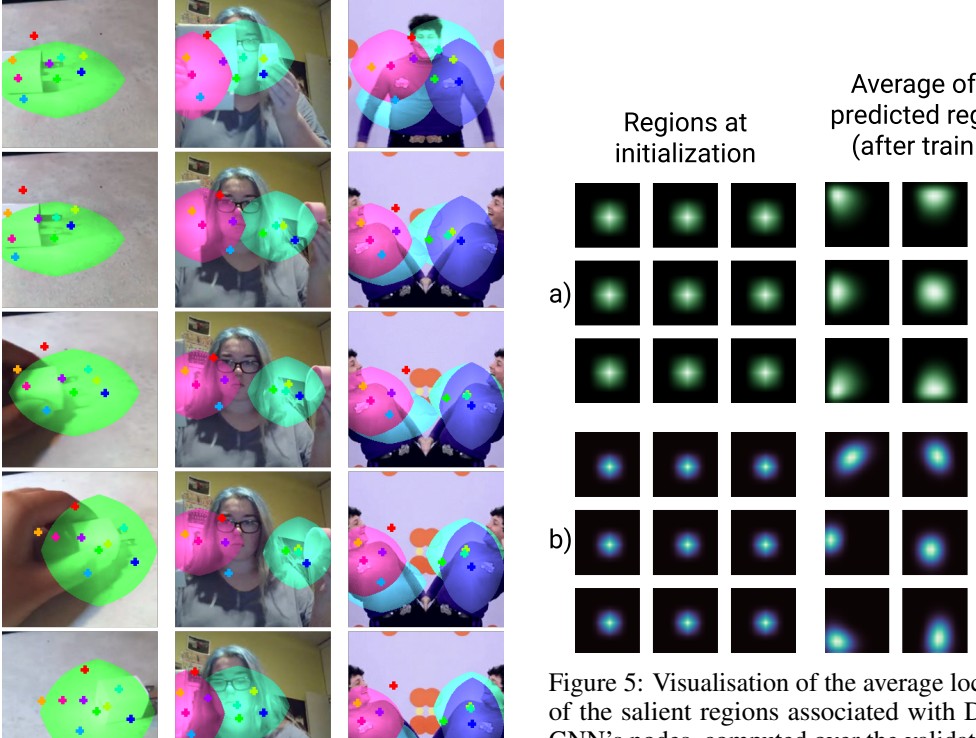

Figure 4: Visualisations of salient regions associated with each node, as predicted by our DyReg-GNN model on videos from Smt-Smt-v2 dataset (Left and Center) and on out-of-distribution real-world videos (Right). Each node learns to move to different relevant regions in the input. For each video, we show the centers corresponding to all the nodes and, for a better visualisation, a subset of the predicted regions. Due to the receptive field of the backbone, the nodes are actually influenced by larger regions in the initial input.

Figure 5: Visualisation of the average locations of the salient regions associated with DyReg-GNN's nodes, computed over the validation set of (a) MultiSyncMNIST and (b) Smt-Smt-v2. In these visualisations the order of the regions is manually selected. In the left column we show the regions at initialisation, and in the right column we present the mean regions as predicted by our learned DyReg-GNN model. Here, we keep the size of the regions fixed, each node has a preferred location in space and assigns salient regions around it. This behaviour is learned by the model to break the symmetry of the nodes.

By initialising $\gamma = 0$ we obtain $h = h_{init}$, $w = w_{init}$ and $\Delta y = \Delta x = 0$. This means that all regions are initialized centered in the reference point $C$ and start with the predefined size. By default we set $h_{init} = \frac{H}{6}$, $w_{init} = \frac{W}{6}$.

## D  Visualising the nodes' regions

The region associated with each node is clearly delimited in space and we can easily visualize them. We train a model on Something-Something-V2 dataset of human-object interactions and in Figure.4 we show its predicted nodes' regions for two videos from the dataset and one out-of-distribution video. Generally the nodes follow relevant regions in the input. We note that the visualisation of the regions is only an approximation of the actual regions that send information to the graph nodes. Each node pools info from a low-resolution region in the latent convolutional features, that corresponds to the high-resolution visualized region. But, the actual area that contributes to each node is actually larger, due to the receptive field of the convolutional network. Moreover, the backbone also contains temporal processing (e.g. in the form of temporal feature shifting in the case of TSM or 3D conv for I3D) such that each node receives information from adjacent time steps. Thus, we expect some misalignment in the visualizations both in space and time.

To better understand how each node attends to the input, we compute the average of its associated regions over the entire evaluation dataset (see Figure. 5). We observe that the regions are initialized in the center of the image and, after training, each node learns to attend to regions around a specific location. For each video, a node predicts a different region, according to the input, but it is situated mostly around a certain part of the image. This behaviour is learned by the model to break the symmetry of the nodes and be able to create an implicit matching between relevant parts of the input and the nodes.

# E   Synthetic Setting

## E.1   Dataset details

Based on [3] we create MultiSyncMNIST. It consists of 10 frames videos of size $128 \times 128$, where MNIST digits move on a black background. Each video has 5 moving digits and a subset of them moves synchronously. Different from the original version, each video could contain multiple instances of the same digit class and any subset can move in the same way. This is done to make it more difficult to distinguish between multiple visual instances. The goal is to detect the smallest and largest digit class among the subset of synchronous digits with each pair of two digits forming a label. In total, we have 55 possible pairs of two digits, and adding a class for videos without synchronous digits results in a 56-way classification task. For example, if a video contains the digits: $\{2, 4, 6, 7, 7\}$ and the subset $\{4, 6, 7\}$ is moving in the same way, it has the label associated with the pair: $\{4, 7\}$. The dataset contains 600k training videos and 10k validation videos.

## E.2   Ablation: Number of Nodes

We investigate the effect on the performance of the number of nodes for different environments, of varying difficulty. We conduct experiments varying the complexity of MultiSyncMNIST dataset, by changing the number of moving digit ($D \in \{3, 5, 9\}$). As expected, we observe that for good performance, it is necessary to set a number of nodes that exceeds the number of relevant entities in the scene.

## E.3   Ablation: Regions' Size

In this subsection, we conduct experiments to investigate the effect of the size of the node regions on the final performance. Each node pools information from latent convolutional features of size $H \times W = 16 \times 16$. We fix the size of each region to $\frac{H}{\lambda}$ where $H = 16$ and $\lambda \in \{6, 7, 8, 11, 16\}$ and show the results of the corresponding models in Table 11. Setting $\lambda = 8$ corresponds to regions having approximately the expected values of the regions predicted by the full DyReg-GNN model. We note that the model is relatively robust to reasonable choices of size but the best performance is achieved when the size of each region is dynamically predicted from the input. We also note that by setting $\lambda = H = 16$ we arrive at the standard bilinear interpolation kernel. This setting leads us to a model that is more unstable in training than the others and obtains poorer results. There are two probable reasons for this. First, the regions cover a small area thus they must be more precise to cover small entities while also being unable to cover large entities in their entirely. Second, the gradients used to update the region parameters are noisier for small regions. This is because, the gradients of the offsets depend on the features of the predicted regions, and for gradients of the offsets to be informative it means that the features in the regions should also be relevant for the final prediction. Smaller regions have a smaller chance of achieving this.

## E.4   Ablation: Comparison to Keypoints Extractor

We conduct an experiment to compare our dynamic way of generating nodes with a previous method [55] that detects keypoints from images. For a fair comparison, we replace the part in our model that predicts the locations of the node regions with a method similar their encoder. The rest of our model would remain the same and will be learned in the same way by the video classification loss. We chose the architecture such that the number of parameters remains the same. As in the original paper, this method predicts the positions but keeps the shape fixed, thus we compare it with our module that has fixed shape, although it obtains poorer results than the full model. We report the results in Table 10.

Table 7: Results on MultiSyncMNIST when varying the number of nodes on datasets of different complexity (with increasing number of moving digits). It is crucial that the number of nodes exceeds the number of important entities in the scene. In the bottom of the table, we also show two additional baselines with the same number of parameters.

| Model | Dataset | | |
|---|---|---|---|
| # digits(D) | D=3 | D=5 | D=9 |
| DyReg 5 Nodes | 98.2 | 89.6 | 64.4 |
| DyReg 9 Nodes | 98.1 | 95.1 | 79.3 |
| DyReg 16 Nodes | **98.4** | **95.6** | **83.0** |
| R18 + Conv-LSTM | - | 89.1 | 50.5 |
| R18 + NL | - | 93.2 | 67.5 |

Table 8: Results on Smt-Smt-V2 val. set, using a single $224 \times 224$ central crop. We observe that DyReg-GNN models improve over the TSM backbone and that it is important to have the kick-start given by the distillation to learn multiple dynamic graph modules.

| Model | Top 1 | Top 5 |
|---|---|---|
| TSM | 61.1 | 86.5 |
| DyReg-GNN r3-4-5 | 62.1 | 87.4 |
| DyReg-GNN r3-4-5 Distill | 62.8 | 87.7 |

Table 9: Ablation on MultiSyncMNIST for showing the importance of recurrence for predicting the regions.

| Model | Accuracy |
|---|---|
| DyReg-GNN without GRU | 91.91 |
| DyReg-GNN | 95.09 |

Table 10: Comparison to Keypoints-based method on MultiSyncMNIST.

| Model | Acc |
|---|---|
| ResNet-18 | 52.29 |
| Fixed | 78.85 |
| Keypoints | 90.60 |
| DyReg-GNN - Pos-Only | 93.41 |
| DyReg-GNN | **95.09** |

Note that the dynamic regions obtained using the keypoints method (denoted as Keypoints) improve over the fixed-regions approach, reinforcing the idea that dynamic regions are helpful for relational processing. However, our DyReg-GNN models obtain better results both when the size of the regions is fixed and especially when the size is also predicted.

## E.5  Ablation: Importance of Recurrence for Region Generation

We conduct an experiment (Table 9) on the MultiSyncMNIST dataset, where we omit the GRU from Eq. 3, thus predicting the regions at each time step only from the features of frame. The performance drops from (DyReg-GNN) to (DyReg-GNN without GRU). This experiment suggests that the temporal modeling in region generation is important for good performance.

## E.6  Ablation: Ground-Truth Boxes

To evaluate the quality of our proposed regions on MultiSyncMNIST, we train our model using ground-truth (gt.) boxes instead of generated regions. As this task is defined by the exact movements of digits, the gt. boxes represents the ideal regions for the relational model, giving an upper bound for our method. This oracle model obtains $97.30\%$ accuracy, while the DyReg-GNN model obtains $95.09\%$. Comparing to the other baselines in the main paper, our DyReg-GNN model obtains closer results to the oracle model, proving the utility of the node generation.

## E.7  Comparison to other baselines

We compare our method to additional baselines by replacing our entire DyReg-GNN module with two other models, as seen in Table 7. The first baseline (R18+Conv-LSTM) consists in a convolutional encoder that reduces the spatial dimensions, a shared LSTM applied independently on each spatial position followed by a convolutional decoder. The second baseline (R18-NL) consists in a Non-Local[30] network. Both modules are applied over the same ResNet 18 backbone and have the same number of parameters as DyReg-GNN. DyReg-GNN surpasses the other baselines and the difference in performance is more significant in the hardest setting.

Table 11: Experiments on MultiSyncMNIST investigating the size of the learned regions. The best performance is obtained when the size is dynamically predicted while the worst is given by a model with the regions kept at the minimum value, corresponding to the standard bilinear interpolation kernel.

| Learnable (Full) | Fix $\lambda = 6$ | Fix $\lambda = 7$ | Fix $\lambda = 8$ | Fix $\lambda = 11$ | Fix bilinear |
|---|---|---|---|---|---|
| 95.09 | 93.41 | 94.11 | 94.04 | 94.03 | 90.99 |

Table 12: Comparison in terms of the number of operations and parameters for a single video of size $224 \times 224$. Comparing to assigning nodes to boxes from external detectors (as in I3D+NL+GCN), our module has a smaller computational overhead.

| Model | Frames | FLOPS | Params |
|---|---|---|---|
| I3D [63] | 32 | 153.0G | 28.0M |
| I3D+NL [30] | 32 | 168.0G | 35.3M |
| I3D+NL+GCN [7] | 32 | 303.0G | 62.2M |
| STM [80] | 16 | 66.5G | 24.0M |
| TSM [67] | 16 | 65.8G | 23.9M |
| TSM + Fixed GNN r4 | 16 | 66.3G | 24.9M |
| TSM + DyReg-GNN r4 | 16 | 66.4G | 25.7M |
| TSM + Fixed r3-4-5 | 16 | 67.2G | 26.1M |
| TSM + DyReg-GNN r3-4-5 | 16 | 67.4G | 28.7M |

# F  Human-Object Interactions

## F.1  Distillation for kick-starting the optimisation

When training models with multiple DyReg-GNN modules, we observe that only the regions of the last module behaves well, thus a single graph module is effectively used. To alleviate this problem we train models with a single graph at different stages, and use their region predictions to distill the larger model, for the first $10\%$ of the training iterations. This kick-starts the learning of all graph modules, improving the overall results, as seen in Table 8.

## F.2  Implementation Details for Detector Experiment

In the main paper, for the comparison with the regions extracted using object detectors (Section 4.1), we use a Faster RCNN ResNet-50 FPN detector[3] pre-trained on MSCOCO dataset. We extract the top-9 detected boxes based on the confidence score and temporally match them using the hungarian algorithm to maximize the IoU between boxes at consecutive time steps.

## F.3  Object-centric metric

To quantify to what degree the nodes cover existing ground-truth objects in the scene, we propose the following metric. We measure the distance between the center of the predicted regions and the center of the gt. objects. For each node region in each frame, we compute the minimum $L_2$ distance to all gt. object bounding boxes and average all of them.

$$Dist_p = \frac{1}{NF} \sum_{f=1}^{F} \sum_{i=1}^{N} min_j |C_i + \Delta_i - B_j|_2 \qquad (15)$$

Vice versa we compute for each gt. box the minimum $L_2$ distance to all predicted regions and average all of them.

$$Dist_r = \frac{1}{N_B F} \sum_{f=1}^{F} \sum_{j=1}^{N_B} min_i |C_i + \Delta_i - B_j|_2 \qquad (16)$$

---

[3]`https://github.com/facebookresearch/detectron2`

In the previous equations, $F$ is the number of frames in the whole dataset, $N$ the number of nodes, $N_B$ the number of objects in the current frame, $C_i + \Delta_i$ is the center of $i$-th node's region and $B_j$ the center of the $j$-th object in the current frame and we average over the whole dataset.

The first score (representing precision) ensures that all the predicted regions are close to real objects, while the second (recall) ensures that all the objects are close to at least one predicted region. To balance them, we present as our final score their harmonic mean.

### F.4 Runtime Analysis

We compute the number of operations, measured in FLOPS, the parameters and the inference time for our model. We evaluate videos of size $224 \times 224$ in batches of 16 on a single NVIDIA GTX 1080 Ti GPU. TSM backbone, TSM + DyReg-GNN-r4, and DyReg-GNN-r3-4-5 run at 35.7, 34.8, and 32.7 videos per second respectively, showing that our DyReg-GNN module does not add a large overhead over the backbone. In Table 12, we compare in terms of number of parameters and operations against other current standard models used in video processing. Note that the I3D-based models uses 32 frames but for our method, the number of operations increases linearly with the number of frames so it is easy to make a fair comparison. The I3D+NL+GCN model counts also the parameters and the operations of the detector module used to extract object boxes. This is characteristic to all the relational models where the nodes are extracted using object detectors. Contrary to this approach, our method has a smaller total complexity by directly predicting salient regions instead of using precise object proposals given by external models.