# OpenReview forum: "Discovering Dynamic Salient Regions for Spatio-Temporal Graph Neural Networks"
_NeurIPS.cc/2021/Conference — NeurIPS 2021 Poster_

### Official Review · Reviewer_9uPY · 2021-07-12

**Rating:** 7
**Confidence:** 4

**Summary:**

This paper proposes a graph based network for spatio-temporal understanding. The core idea of the paper is to learn to select visual features for node representations in graph dynamically using convolutional encoders. The paper shows competitive results in the Smth-Smth v1 and v2 datasets and MultiSync MNIST. The contributions of the paper are demonstrating a method about how to learn to select dynamic regions and making it work in the context of graph neural networks.

**Limitations And Societal Impact:**

The paper adequately addresses the social impact.

**Main Review:**

The paper addresses an important problem of spatio-temporal understanding. It is important to investigate approaches on understanding this modality more deeply (e.g. via reasoning) rather than superficially building deeper networks and this paper makes a step in the right direction.

I have the following comments and questions about the paper:

1) What are the benefit of using dynamic conv features apart from performance scores? The benefit of using static nodes (e.g. using objects) is that there is increased model interpretability and the model learns to establish between static entities in the dataset. Using convolutional features that change every time-step based on output from conv layers is not very interpretable. It may increase the performance of your model, since the model learns to attend to larger number of foreground and background features. However, this benefit is also seen with object detectors where ROIs are used [A].

2) There is criticism of datasets in Section 4, some of which is valid. Kinetics does have a high spatial bias but does contain fine-grained classes (e.g. around basketball) that are far from being solved and it would be interesting to see how your model can separate actions for classes that involve similar objects (or salient regions). Similarly, the size of CATER was limited by design. Also, [68] does release an updated dataset that does require longer term reasoning and maybe worth looking into. CATER is perhaps a great fit for the central idea of the paper as the attention region should dynamically move from snitch to cones (in Task 3) as the video progresses.

3) In the contributions, you mention that the paper “enhances relational processing”, it is not clear that the relational processing enhaced i.e. better higher order relations are established. My intuition is that by regressing and attending to a large number of region features the model is better regularized as well as gets additional signals that a pre-trained object detector misses.

4) How did you pick 9 regions/objects? Perhaps to make the system completely dynamic, this can be a learned parameter as well.

After reviewing rebuttal responses from authors to my review and other reviews, I decided to change my score to Accept.

[A] Attend and Interact: Higher-Order Object Interactions for Video Understanding https://arxiv.org/abs/1711.06330

**Time Spent Reviewing:**

5 hours

---

> ### Author Response · Authors · 2021-08-10
> **Response to Reviewer 9uPY**
>
> We are thankful that the reviewer appreciates our work and believes that the novel direction we propose is a step in the right direction. The reviewer is not certain about the advantages of using dynamic regions for enhancing relational processing. We prove that our model *learns relevant entities* (predicted regions are correlated to objects), so our relational module processes interaction between meaningful latent entities. Moreover, we show that *more dynamic regions lead to better relational processing*, as we clearly validate in the ablation studies. We definitely prove that our method brings *important benefits* over the methods based on ROI object detectors in both performance and efficiency.
>
> In the following, we discuss in more detail each point raised by the reviewer. We will integrate all the feedback into the final version of the paper.
>
> ### Regarding the advantages compared to ROI-based
>
> Compared to pure convolutional methods, our model offers a higher degree of interpretability obtained by visualizing the predicted regions that are used in the relational processing. Compared to ROI-based models extracted using object detectors, our model *does not require object-level supervision*. Our model is *more flexible*, and it could learn from the final video-level supervision to produce regions that are *more aligned* with the current tasks. Moreover, as shown in Table 1, DyReg-GNN is much *more efficient* than object detectors methods. The module using an object detector adds 41.1 GFLOPS, while our DyReG-GNN module adds only 1.6 GFLOPs. We will include [A] in the related work.
>
>
> ### Regarding the improved relational processing.
>
> As the reviewer notes, it is very important to determine if the improvements come from better modeling of the relations or from other causes like better optimization. We have multiple evidence suggesting that our method offers better relational modeling:
>
> First, GNN models are designed to model the relations in the data, and we show that our method makes them capture such relations more easily by focusing only on relevant regions in the input. We have a quantitative evaluation of the regions predicted by our model and prove that they *correlate with true objects*. Thus, compared to the fixed regions approach our method processes relations between more relevant entities. Therefore we argue that the obtained relations are more relevant, and we say that our approach improves the relational reasoning.
>
> Second, the synthetic dataset Multi SyncMNIST is *specially designed* to measure the relational capabilities of a model. The goal is to find the digits that are related by their movements (they move synchronously). Similarly, the Something-Something dataset is designed to have classes that represent relations between 2 or multiple entities (Holding something behind something, Moving something and something so they collide with each other, Moving something and something so they pass each other). Solving both of these two datasets requires relational reasoning.
>
> Although improvements could come from different aspects, such as better regularisation, we believe that our ablation studies (Table 5) clearly emphasize that our model improves relational reasoning. The full model obtains the best results although it has the same supervision, attends to regions of similar size and has the same degree of implicit and explicit regularisation. Thus the effect of better supervision and optimization and attendance to larger regions are not crucial for the model's performance.
>
> ### Regarding the number of nodes
>
> We choose the number of nodes such that it is larger than the number of entities in any scene. In Appendix E.2, we validate that the model is *robust* to a larger number of nodes and obtains the best performance when the number of nodes exceeds the number of relevant entities in the scene.
> We agree with the reviewer that predicting the number of nodes is a good idea and has the potential to improve the performance further. We are considering this direction for future work because it needs completely new, non-trivial mechanisms to make the number of nodes learnable.

---

### Official Review · Reviewer_uuaS · 2021-07-15

**Rating:** 4
**Confidence:** 3

**Summary:**

The paper proposes to improve spatio-temporal graph neural networks by adaptively computing the input node features from spatially localized attention regions. The attention regions are dynamically predicted from each input frame, which is in contrast to previous methods that use fixed regions or pretrained object detectors. It is shown that by only optimizing a video classification loss, the model can learn to attend to objects, thereby extracting object-centric node features and facilitating interaction modeling. On the Something-Something-V1&V2 datasets, the model achieves comparable or slightly better classification accuracy than previous work, and can consistently boost performance when combined with multiple backbones.

**Limitations And Societal Impact:**

In addition to the limitations mentioned in the paper, I think the proposed model may not have good temporal consistency when objects move long distances, and thus the improvement in interaction modeling can be limited in long videos.

**Main Review:**

It is an interesting idea to let the nodes of a graph net dynamically attend to relevant input regions. This has the potential for better interpretability and interaction modeling. However, similar ideas have been proposed in previous methods like OP3[1] and V-CDN[2], where they predict object masks / keypoints and use graph nets to model dynamics, all without supervision. Compared to these methods, the strength of this paper seems to be that it can work on real-world datasets like Something-Something-V1&V2, when given the video class labels. However, I had some concern about the experiments:

- The main metric considered is the classification accuracy, on which using dynamically adapted regions only slightly outperforms using fixed regions (Table 1). Also, in Appendix Figure 5, it seems that each node covers a specific region. These results lead to the question: Is it necessary to use dynamic regions?
- For object-centric representations to benefit interaction modeling, it is best if the representations are temporally consistent, i.e., same node capturing same object across time. However, I did not see much evalution on this temporal consistency aspect, and Figure 5 seems to suggest that if an object moves a long distance, it will be captured by different nodes.
- I was a bit confused about the visualization of node regions in Figures 3 and 4. In particular, the number of regions in Figure 4 is much smaller than the number of nodes, so what nodes are visualized? Also the learned position of nodes looks similar to keypoints [2-4], which can be predicted without supervision even for realistic images. I think it would be better to have some discussion about these methods.

**MINOR COMMENTS**

- What is the benefit of the proposed kernel function compared to spatial transformer networks?

[1] Entity Abstraction in Visual Model-Based Reinforcement Learning (CoRL 2019).

[2] Causal Discovery in Physical Systems from Videos (NeurIPS 2020).

[3] Unsupervised Learning of Object Landmarks through Conditional Image Generation (NeurIPS 2018).

[4] Unsupervised Learning of Object Keypoints for Perception and Control (NeurIPS 2019).

**Time Spent Reviewing:**

5

---

> ### Author Response · Authors · 2021-08-10
> **Response to Reviewer uuaS**
>
> We appreciate that the reviewer finds our work interesting, offering a higher degree of interpretability. The reviewer raised concerns regarding the need to use dynamic regions for modeling complex interactions. The necessity of dynamic regions is *the core idea* of our paper and we proved several times that the predicted regions are both relevant on their own (correlates with the objects) and also improve the overall performance. The ablations validate that *the more dynamic the regions, the higher the performance*.
>
> We propose a method with connection to multiple domains, being the first to show the benefits of using predicted salient regions in a relational model capable of increasing the performance in real-world scenarios. We think that our work is a good stepping stone towards a direction that combines the advances made in several adjacent sub-domains such as: salient region/keypoints discovery, object-centric models and relational processing with GNNs.
>
> In the following, we discuss each remark raised by the reviewer, and we will integrate all the feedback and related works into the final version of the paper.
>
> ### The necessity of dynamic regions
>
> In our paper, we perform extensive analyses that show the necessity of using dynamic regions for relational modeling. As the reviewer noted, our method consistently improves the performance of multiple backbones on different datasets, and we showed that dynamic regions play a crucial role.
>
> As the other reviewers observe, we perform thorough ablation studies on the challenging synthetic setting where we clearly establish that dynamic regions obtain large performance gains (Fixed grid $78.85$ vs $95.09$ DyReg-GNN). Table 5 shows that fixed regions obtain the poorest results, and the performance consistently increases each time we allow the model to be more dynamic. The final Dyreg-GNN (95.09) improves over fixed regions (78.85), over regions optimized for the dataset (81.48), over regions predicted but constant in time (86.77) and over predicted regions at each time step, with constant shape (93.41). This proves that we need to have dynamic regions that are predicted based on the current input and change both their position and shape at each time step.
>
> Figure 5 shows the average position of the regions taken over the entire dataset. The regions tend to move around a specific location. But as measured by the correlation with true objects, and as seen in the qualitative examples, the regions generally move such that they cover salient regions.
>
> It is crucial to have a mechanism that decides how to assign each entity in the image to one of the nodes in a consistent way, such that the learning is stable. This problem of binding an object to a node is challenging [A], and one solution would be to make the assignment based on spatial position. For each video, a node predicts a different region, according to the input, but it is situated mostly around a certain part of the image. This behaviour is *learned* by the model (without any constraint for this) to break the symmetry of the nodes and be able to create an implicit matching between the relevant parts of the input and the nodes.
>
> *[A] Greff, van Steenkiste, Schmidhuber (2020). On the binding problem in artificial neural networks.*
>
> ### Temporal consistency
>
> The reviewer correctly observed that for long distances, an object could be captured by different nodes. To quantify to what degree this is happening, we compute a metric that shows how many times a node switches from covering an object to another. We evaluate this consistency on the MultiSyncMNIST dataset since for the Smt-Smt dataset, we do not have the trajectories of GT objects. We match each region with the closest digit (in L2 distance) and, for each node, we count how many times the associated digit from time step $t$ is different from the associated digit from time step $t+1$. For our method that does not explicitly enforce temporal consistency, each node switches 1.17 times on average. While temporal consistency has the potential to improve our method further, it is a non-trivial problem that we plan to address in future work.
>
> ### Visualization
>
> In Figure 4, we show the centers of all the nodes but show the entire region only for a subset of nodes for better visualization.
>
> ### Keypoints
>
> We thank the reviewer for pointing out the connection to keypoint detection methods. They are indeed relevant, and we will adequately discuss them in the related work.
> From the mentioned papers, the most relevant to our work is [3] as it is the only one that works on real-world videos. Their goal is specifically to learn keypoints, while ours is to improve relational processing. This model learns unsupervised keypoints from a video reconstruction loss, while we learn salient region coordinates from the loss of the task of interest (video classification).
>
> We argue that our approach leads to regions that are *more relevant to the current task*, as they are optimized for it, they are not just important for reconstructing the image. Different from keypoints approaches, our method could learn to ignore distinctive regions in the input (essential for reconstruction) but that are not relevant for the classification task.
>
> A downside of these methods [3,4] is that they are not designed to work with *multiple instances* of the same kind of object. E.g. If the scene contains two similar objects at two different positions, the convolutional map $S$ that estimates the probabilities of a keypoint will highlight both positions almost equally ($S(u_1) == S(u_2)$). The keypoint location is predicted as the mean position of the activations: $u^{keypoint} = \sum_{u \in positions} u S(u)$ (Eq. 1 from [3]). So instead of identifying the two objects, the method will predict the position located in the middle of them.
>
> We could freeze the entire model [3] and use it to predict the centers of our regions, but this has a couple of downsides: 1) the keypoints are not aligned with the current classification task, 2) it requires training of this additional method and 3) it does not predict the size of the regions (that we showed to be an essential aspect - Table 5).
>
> Alternatively, we could replace the part in our model that predicts the locations of our node regions with a method similar to the encoder part of [3], that predicts the keypoints. The rest of our model would remain the same and will be learned in the same way by the video classification loss. We test this idea by replacing the part of our model that predicts the region's offsets (Eq. 1 - Eq. 4) with a module similar to the encoder of [3] that predicts the keypoints coordinates. We chose the hyperparameters such that the number of parameters remains the same. As in the original paper [3], this method predicts the positions but keeps the shape fixed, thus we compare it with our module that has fixed shape, although it obtains poorer results than the full model.
>
>
> | Model   |      Accuracy      |
> |:----------:|:-------------:|
> | Fixed Grid |  78.85 |
> | Keypoints - Dyreg-GNN  |   90.60  |
> | Dyreg-GNN Position - Only (fixed size) |  93.41 |
> | DyReg-GNN |   **95.09**  |
>
>
> Note that the dynamic regions obtained using the keypoints method ([3]) improve over the fixed-regions approach, reinforcing the idea that *dynamic regions are helpful* for relational processing. However, our DyReg-GNN models obtain better results both when the size of the regions is fixed and especially when the size is also predicted.
> We will have a discussion about the connections to all papers mentioned by the reviewer, and we will also add the comparison to [3] in the experimental section.
>
> ### STN
>
> As we said in the related work and as R1(y7mA) also observed, our work is related to STN as we use a similar interpolation for pooling. One advantage of our method is that each node independently pools info from different regions, while in STN, all the positions interpolate the input according to the same global transformation. Also, instead of bilinear interpolation that combines the 4 closest points, we interpolate from larger regions (according to the predicted size) and show that this aspect is important for the optimization (Appendix E.3.)

---

### Official Review · Reviewer_5S3R · 2021-07-16

**Rating:** 5
**Confidence:** 3

**Summary:**

This paper introduces a graph-based model for salient region discovery in videos, called Spatio-Temporal Graph Neural Networks (DyReg-GNN). The key idea is to capture latent interactions between various entities by graph; moreover, RNN is used to capture inter-flame information. Generally, the idea is reasonable and the experiments show the effectiveness of the proposed approach.


**Limitations And Societal Impact:**

- It is unclear to me why the proposed method is unsupervised. It seems that the proposed approach requires video-class level supervision for learning parameters;

- The graph net/CNNs for better feature representation learning has been studied by many other research work. Even though the paper is the first to introduce it for salient region discovery, the contributions of the paper are still limited as a NeurIPS paper.

- The experiments show that the proposed approach achieves competitive results; yet, it dose not outperform SOTA approaches such as MSNet.

**Main Review:**

+ The idea of using graph-based techniques for capturing inter-entities is reasonable;
+ The paper describes each component clearly.

**Time Spent Reviewing:**

2 hours

---

> ### Author Response · Authors · 2021-08-10
> **Response to Reviewer 5S3R**
>
> We highlight that predicting visual nodes from videos, useful for modeling interactions, is an important contribution of this work that is not explored in the literature. Current approaches are exclusively based on *grid-like nodes* with limited capabilities or entities extracted using *object-detectors* that are both inefficient and require heavy training and annotations. Our paper proposes a novel method that relies on modeling the relation between salient regions and validates the idea that *dynamic nodes are essential* for relational reasoning. This is an important contribution both for the novelty of the method and for validating an important direction that *opens the door for further improvements*.
>
> We discuss below the main concerns raised by the reviewer and we will integrate all the feedback into the final version of the paper.
>
> ### Unsupervised method.
>
> Our method is supervised from the video label. An advantage of the method is that it does not require supervision *at the object level*, it does not require object boxes, segmentations or any other annotations at this level. This is what we mean when we say that it is *unsupervised at the region level*. This is in contrast to methods that need object detectors, that are trained with *supervision from object boxes*, and is an important contribution of our method.
>
> ### Contributions
> In the main paper, we give an extensive related work, where we clearly position our method in the current literature and make connections to multiple adjacent sub-domains. The reviewer identifies our work as novel but raises concerns about the relevance of the contributions.
>
> There are multiple methods involving GNNs in vision, but they all have the downside of using nodes from object detectors or fixed regions. Our method gives a solution to this limitation and emphasizes a direction to be further improved, that of *dynamically predicting salient regions used in relational processing*. As the other reviewers noticed, “this paper makes a step in the right direction.”(R4(9uPY)), “explores a novel direction”(R1(y7mA)), “addressing an important problem of spatio-temporal understanding.”(R4(9uPY)) by tackling “a crucial limitation of spatio-temporal GNN” (R1(y7mA)) and having “the potential for better interpretability and interaction modeling” (R3(uuaS)).
>
> ### SOTA
> As we mentioned, we tackle an underexplored problem in graph-based video processing. Our paper validates the necessity of dynamic nodes in *multiple controlled experiments* and gives an important direction to be followed. We show improvement for *multiple datasets and backbones*, proving the effectiveness of the method. Our approach obtains the best performance among the *graph-based methods*, as seen in Table 3.

---

### Official Review · Reviewer_y7mA · 2021-07-16

**Rating:** 7
**Confidence:** 5

**Summary:**

The paper proposes a novel approach for generating dynamic visual nodes, ideally capturing salient visual regions, which can then be fed to a spatio-temporal Graph Neural Network in order to model intra-region relationships. In contrast to prior work, which uses feature map columns in a grid or object detector bounding boxes as the input visual nodes to a GNN, the proposed DyReg-GNN learns to localize a fixed number of visual nodes from a video, without using object detectors or assuming spatial annotations.

To achieve this, the authors propose using node-specific trainable neural networks, which output region locations and sizes based on the input spatio-temporal feature map. Then spatio-temporal features are differentiably-pooled from each one of these regions, are fed to a standard GNN, are refined based on interactions, and finally are mapped back to their original spatio-temporal positions to yield a refined spatial feature map. Therefore, the DyReg-GNN can be inserted at any intermediate level in a standard convolutional model.

Results show that the proposed approach significantly outperforms GNNs that are applied on a fixed grid on a synthetic dataset (MultiSync MNIST), is much more efficient than methods that use object detectors, and the generated regions are qualitatively shown to correlate with objects, change size over time, and even retain object identities over time. The method is also evaluated on Sth-Sth V1, where it leads to a competitive performance, although the improvement over the grid baseline is much smaller.

**Limitations And Societal Impact:**

Yes.

**Main Review:**

Strengths
=======

S1) Well-motivated approach: The paper addresses a crucial limitation of spatio-temporal GNN approaches applied to video, namely the need for object detectors in order to generate candidate salient regions. This is not only computationally expensive, but is also limited by the proposal recall of object detectors. The proposed approach aims to remove the need for this step, by automatically learning to localize a fixed number of salient regions in each video frame, without spatial annotations.

S2) Original framework: The approach is very nicely positioned w.r.t the prior work and is exploring a novel direction, that of automatically generating salient regions beyond grids. It cleverly leverages networks that output location and size, differentiable pooling ideas (such as the ones in Spatial Transformer Networks[39]), and spatio-temporal GNNs[3].

S3) Well-designed ablations in a synthetic dataset: Except for tackling a novel direction, another strong point of the paper are the experiments on the MultiSyncMNIST dataset. These controlled experiments clearly demonstrate: a) that using the discovered region nodes instead of a fixed grid improves video classification (Table 5), b) using the discovered, visually grounded, region nodes is better than finding latent semantic nodes and applying the GNN on them (Table 6), c) generated nodes tend to follow the digits locations, while also adapting region's size.

S4) The paper is very well-written, is easy to read and equations are properly explained. The approach also seems technically sound.

S5) The authors have provided rich qualitative examples, videos, implementation details, and code. A lot of limitations are also discussed.

Weaknesses
==========
W1) Choice of benchmark dataset (Something-Something): I think that the proposed method would be useful in real-life applications if it could significantly improve upon a fixed grid approach in real datasets, without the added overhead of an object detector. I appreciate that the authors compared these 3 settings. However, I feel that the chosen real-life dataset is too limited in terms of the richness of scenes. This does not only limit the expected impact from using salient regions instead of grids, but might also be an easy dataset for the unsupervised region discovery.For example, most scenes have only a few objects and a single person (hands) interacting with them. Therefore, although action recognition in SthSth is indeed a challenging spatio-temporal reasoning task, finding salient regions is not as hard, since most of the foreground blobs are relevant. We can see that from the experimental results as well. In Table 1, we observe that using a fixed grid vs an object detector vs the proposed salient nodes yields an accuracy of 64.1 vs 64.0 vs 64.8. The fact that object detections do not improve over the grid might be because the grid already captures all the relevant objects and there are not many irrelevant grid cells.

W2) Missing ablations: Was there an ablation about not using the GRU when generating nodes over time (Eq. 3)? Also, what is the impact of the attention component in Eq. 8? Could it be that grid-based GNNs without attention would be much worse than the proposed DyReg-GNN? Did the authors experiment with different number of GNN layers or different loss scalar weight \lambda?

W3) Evaluation of alignment of generated regions with ground-truth objects: Why (only) use the L2 distance for comparing regions with ground-truth object regions, and not IoU as well (Table 1)? Also, an additional reported metric could be the number of salient regions covering each object instance. The authors acknowledged this limitation, namely that because of the fixed number of nodes, a lot of the generated regions can focus on the same object. Object recall is also an important metric. It would be also interesting to have a quantitative evaluation on whether the model learns some form of tracking, although it is not the primary goal.

W4) Comparison with SOTA: The approach improves marginally improved results compared to RSTG (same? GNN applied on feature grids) - 49.2 vs 49.9%. Also, the method is not state-of-the-art when compared to non-graph methods that use the same backbone and number of frames.  Question: does the RSTG[3] also use the TSM module?

Minor comments
============
DC1) The authors could also cite: Dynamic Graph Modules for. Modeling Object-Object Interactions in. Activity Recognition, BMCV19 , which shares the same ideas of having a latent, location graph, that is visually grounded, although that method also uses object detections for the visual graph.
DC2) In the state-of-the-art tables, it would be useful to add a column regarding the nodes used by the compared GNNs (grid vs object vs dynamic).
DC3) Since the node position generation networks are 9 fixed networks, was there any qualitative analysis on what types of regions they generate? For example, is there any correlation between the region generated by the first node generation network for similar scenes? Is there any cluster of objects that arises?

Rating Justification
===============
Overall, the paper proposes a novel approach for spatio-temporal interaction modeling in videos, which tackles the difficult task of dynamically learning salient regions without spatial annotations. The approach is well-motivated, thoughtfully designed and experimental results on synthetic data validate its effectiveness. I am still skeptical whether the approach would be feasible in video datasets with richer scenes, such as Charades or cooking datasets, where object detection is itself more challenging than SthSth. In such datasets, not only the small fixed number of nodes could be an issue, but also training could be harder since the model is ideally trying to capture object detection and object tracking without annotations.  Still, I think the work in its current form opens up an interesting direction in the field and is showing promising results.

**Time Spent Reviewing:**

5

---

> ### Author Response · Authors · 2021-08-10
> **Response to Reviewer y7mA**
>
> We thank the reviewer for the very detailed review and constructive feedback and for correctly identifying and clearly presenting the strengths of our paper. We agree that our work is the first step in a new, exciting direction, and we hope that other methods will follow, extending and improving on the idea of dynamically predicted regions for relational processing.
>
> In the following, we discuss the main remarks pointed out by the reviewer. We will integrate all the suggestions into the final version of the paper.
>
>
> ### W1) Choice of benchmark dataset (Something-Something).
>
> As the reviewer points out, we chose Something-Something dataset as an instance of a challenging spatio-temporal reasoning task, where classes consist of complex actions and interactions. We think that it will be valuable to have an in-depth analysis of the relation between the behaviour of relational models with different node types (extracted from grids or detected objects or dynamically predicted) and the complexity of the scene (expressed both in terms of relations and entities) on different datasets. We leave it as future work.
>
> ### W2) Missing ablations.
>
> As the reviewer suggested, we conduct an experiment on the MultiSyncMNIST dataset, where we omit the GRU from Eq. 3, thus predicting the regions at each time step $t$ only from the features of frame $t$. The performance drops from $95.09$ (full DyReg-GNN) to $91.9$ (DyReg-GNN without GRU). This suggests that the temporal modeling is important for good performance.
>
>
> | Model   |      Accuracy      |
> |:----------:|:-------------:|
> | DyReg-GNN full |  **95.09** |
> | DyReg-GNN without GRU |   91.91  |
>
> In this paper we show that spatio-temporal GNN models are enhanced by nodes extracted from dynamic regions, but we do not focus on the specific relational module. We chose the GNN formulation to be similar to those proposed in prior works, and we have comprehensive validations and ablations only for the node-prediction module.
>
>
> ### W3) Evaluation of alignment of generated regions with ground-truth objects.
>
> **Recall**
>
> As defined in Appendix F.3, we compute for each node the minimum $L_2$ distance to all GT boxes ($L_2\~dist_p$) and vice versa, we compute for each GT object the minimum $L_2$ distance to all regions ($L_2\~dist_r$). In the main paper, we report as $L_2$ distance the harmonic mean between these two metrics. $dist_p$ gives a measure of precision, as it reaches its minimum when each predicted region covers a GT object, while $dist_r$ gives a measure of recall as it reaches its minimum when all GT objects are covered by a region. In the table below, we report all the components of the metric.
>
>
>
> | Model   |   $L_2\~dist_p$    |  $L_2\~dist_r$ | $L_2\~dist$ |
> |:----------:|:-------------:|:-------------:|:-------------:|
> | DyReg-GNN |  0.212 |  **0.093** |  **0.129**  |
> | Detector |   **0.171** | 0.102  |  **0.125** |
> | Grid  |  0.354  |  0.113  | 0.170 |
>
> In terms of recall, we observe that Dyreg-GNN outperforms the supervised detector, having a better $dist_r$. This means that for each GT object, our method has a region that is closer than the boxes given by the detectors.
>
> **IoU**
>
> DyReg-GNN predicts regions in the low-resolution space of the convolutional map, where each point has a large receptive field. It is difficult to find for the convolutional map the correspondence of a boundary in the input image. For a blob in the convolutional map, the center is easier to transpose in the original image, while the exact shape is more challenging to determine. For a predicted region of conv features, their effective receptive field (the pixels in the original input that affect it) could be smaller or larger depending on the content. Having access only to the shape in the latent conv feature space, it's impossible to compute the actual effective receptive field that determines the regions in the input coordinates. This is why we rely on the L2 distance as a quantitative measure for our regions.
>
> **Temporal consistency**
>
> To quantify our method's temporal consistency, we compute a metric that shows how many times a node switches from covering an object to another. We evaluate this consistency on the MultiSyncMNIST dataset since for the Smt-Smt dataset we do not have the trajectories of GT objects. We match each region with the closest digit (in L2 distance) and, for each node, we count how many times the associated digit from time step $t$ is different from the associated digit from time step $t+1$. For our method that does not explicitly enforce temporal consistency), each node switches 1.17 times on average.
>
>
> ### W4)  Comparison with SOTA.
>
> Our GNN formulation differs from RSTG [3] in several points: they use more nodes, multiple scales, each with separate parameters in the message-passing and also have additional update functions in the GNN. RSTG uses I3D as a backbone. We designed the experiment in Table 1 as a completely fair comparison against a model with the same backbone and the same GNN as DyReg-GNN (with a similar recurrent formulation as RSTG, but with differences highlighted above), but different node regions (dynamic vs grid).
>
> ### Minor comments.
>
> **DC1)** We thank the reviewer for the relevant reference, and we will discuss it in the related work and compare against it in the final version of the paper.
>
> **DC3)** It is important to have a mechanism that decides how to assign each entity in the image to one of the nodes in a consistent way, such that the learning is stable. This problem of binding an object to a node is challenging [A], and one solution would be to make the assignment based on spatial position. For each video, a node predicts a different region, according to the input, but it is situated mostly around a certain part of the image (see Appendix D). This behaviour is learned by the model (without any constraint for this) to break the symmetry of the nodes and be able to create an implicit matching between relevant parts of the input and the nodes.
>
> *[A] Greff, van Steenkiste, Schmidhuber (2020). On the binding problem in artificial neural networks.*

---

### Author Response · Authors · 2021-08-10
**General response**

We thank all the reviewers for their insightful comments and valuable suggestions. We respond to each reviewer individually, and we will include in the final version of the paper their suggestions

We are delighted the reviewers agree that we propose a sound method in a *novel, important direction*, offering a good solution for a  *crucial limitation* of video processing with spatio-temporal GNNs (the reliance on detected boxes or fixed regions).

We propose a method with connection to multiple domains, being the first to show the benefits of using predicted salient regions in a relational model that is capable of increasing the performance in real-world scenarios. We think that our work is a good stepping stone towards a direction that combines the advances made in several adjacent sub-domains such as salient region/keypoints discovery, object-centric models and relational processing with GNNs.

### Additional Experiments
As reviewers suggested, we also include two additional experiments: one that validates the importance of recurrence for region prediction and one to compare against an unsupervised keypoint detection method.

1. As R1(y7mA) suggested, we perform an experiment on the MultiSyncMNIST dataset where we omit the GRU from Eq. 3. The performance drops, suggesting that the temporal modeling is essential for good performance.

| Model   |      Accuracy      |
|:----------:|:-------------:|
| DyReg-GNN full |  95.09 |
| DyReg-GNN without GRU |   91.9  |


2. We also replace the module generating the regions' centers with a model suggested by R3 (uuaS), obtaining a model denoted as Keypoints - Dyreg-GNN, but it obtains weaker results than our proposed approach.

| Model   |      Accuracy      |
|:----------:|:-------------:|
| Fixed Grid |  78.85 |
| Keypoints - Dyreg-GNN  |   90.60  |
| Dyreg-GNN Position - Only (fixed size) |  93.41 |
| DyReg-GNN |   95.09  |

\
\
We respond to each reviewer individually in more detail, in separate comments.

---

### Decision · Program_Chairs · 2021-09-27

**Decision:**

Accept (Poster)

**Comment:**

This paper received both positive and negative reviews. After a round of discussion between the reviewers and reading the author's rebuttal, three of the four reviewers recommended accepting the paper and I am happy to accept it. I encourage the authors to include in the final version the additional material that they mentioned in the rebuttal.